# The Effect of Non-Cognitive Ability on Farmer’s Ecological Protection of Farmland: Evidence from Major Tea Producing Areas in China

**DOI:** 10.3390/ijerph19137598

**Published:** 2022-06-21

**Authors:** Xiaohuan Wang, Yifei Ma, Hua Li, Caixia Xue

**Affiliations:** 1College of Economics and Management, Northwest A&F University, Xianyang 712100, China; xiaohuan@nwafu.edu.cn (X.W.); y.ma@rsm.nl (Y.M.); xuecaixia008@nwafu.edu.cn (C.X.); 2Rotterdam School of Management, Erasmus University Rotterdam, 3062 PA Rotterdam, The Netherlands

**Keywords:** ecological protection of farmland, non-cognitive ability, social capital, information channels, value perception

## Abstract

Ecological protection of farmland is an important means to reduce agricultural non-point source pollution and improve the quality of agricultural products. As the main body of current agricultural production and operation, the aging labor force has insufficient cognitive ability and low ability to learn actively, which is not conducive to transforming the green output. However, non-cognitive abilities closely related to the acquired environment can promote the elderly farmers’ farmland ecological protection behavior by improving life satisfaction and social adaptability. Based on the above background, using the survey data of 964 farmers in China, the bivariate Probit model was used to empirically test the influence mechanism of non-cognitive ability on the ecological protection behavior of farmland. The study found that non-cognitive ability significantly promoted farmer’s ecological protection of farmland in China. Specifically, the variables of non-cognitive ability, social communication ability, active learning ability, self-efficacy, stress resistance, altruistic tendency and individual resilience were found to significantly promote ecological protection of farmland. Mechanism analysis showed that non-cognitive ability promoted the ecological protection behavior of farmland by expanding social capital, information channels and improving technical value perception. A heterogeneity analysis revealed that non-cognitive ability had a greater impact on ecological protection behavior of farmland in the elderly and low-income groups. Therefore, government should attach importance to improving farmers’ non-cognitive abilities, further increase technical publicity, and build a communication platform for farmers in order to promote the ecological protection of farmland.

## 1. Introduction

China has been entering a particular period of green production transformation since 2005 [1]. Improving green production at the micro-level has become a significant area concern for both the government and academia. Farmland is an essential foundation for human survival and development. Strengthening the quality protection of farmland is the premise and key to realizing sustainable agricultural development, which is also an inevitable requirement [2]. However, the excessive input of chemical elements has caused a “metabolic fracture” in agriculture that has led to the need to strengthen the protection of farmland quality in China [3]. In the “No. 1 Central Document of China” released in 2021, the central government of China stressed that it should comprehensively promote rural talent and ecological revitalization and give full play to the functions of agriculture as a product supply and ecological conservation. Ecological protection of farmland has become an important measure to build rural revitalization and effectively promote the theory of “two mountains” (The concept was proposed by the Chinese government in 2005. The basic idea is that clear water and lush mountains are invaluable assets).

Schultz 1964 [4] noted that investments in human capital are key to transforming traditional agriculture. According to traditional human capital theory, education level, health status and cognitive ability are important influencing factors of individual behavioral decision-making. However, as the vital force in the current green transformation of Chinese agriculture, the cognitive ability of the middle-aged and elderly labor force and other explicit human capital has shown a decline [5]. Moreover, it is not easy to improve its human capital through training and other channels. This leads to the increasingly prominent contradiction between human capital and agricultural transformation, and restricts the process of agricultural modernization [6]. At the same time, studies have found that when the explicit human capital such as education level, health status, and skills are basically the same, there are still considerable differences in individual behavioral decision-making and results [7], mainly due to non-cognitive ability (NCA), such as individual preferences and attitudes, which play an essential role in this process [8,9]. NCA can positively affect ecological conservation behaviors of farmland by improving the life satisfaction of older workers, relieving depression, and improving their social adaptability [5].

Domestic and foreign economic circles have paid attention to non-cognitive abilities for a long time [10]. In 2001, Heckman et al. [11] put forward a new theoretical framework of human capital with ability as the core. They believed that ability includes both cognitive ability (CA) and non-cognitive ability, and especially emphasized the critical role of NCA. Among them, CA refers to personal thinking, reasoning, memory and other abilities. NCA refers to a relatively stable feeling, thought, and behavior pattern which does not directly participate in cognition but plays a role in the cognitive process. NCA is the individual reaction to a specific situation, measured by the “big five” personality traits [12]. In recent years, exploring the personal effects of NCA from the perspective of personality traits has become a hot topic in labor economics. A study found that non-cognitive abilities were significantly associated with wage levels in the labor market [13], educational participation [14], economic and financial decision-making [15], and corporate decision-making [16]. Bowles et al. [17] found that NCA differences can partially explain the income inequality caused by unobservable factors, and reflect individual ability differences that cannot be reflected by factors such as education and experience. Research shows that NCA, such as individual self-confidence, creativity, sense of achievement, stress resistance and risk appetite significantly promote farmers’ entrepreneurial decision-making and performance [18]. Some studies have shown that the influence of NCA on individual behavior even exceeds the role of traditional human capital such as education and cognitive ability, especially for low-and middle-skilled people [19]. For example, research has found that in a labor market characterized by low skills, non-cognitive abilities such as employee loyalty, obedience and persistence are more valuable to employers than cognitive abilities such as intelligence [10,20]. Wang [21] found that non-cognitive abilities give older people a more significant help in network access. In addition, NCA can give older workers a greater advantage in life satisfaction, alleviate depression, and improve their social resilience, thereby positively affecting farmland ecological protection behavior.

A farmer’s decision with regard to farmland ecological protection involves a complex decision-making process [22]. Over the years, academia has investigated farmland ecological protective technologies, mainly focusing on the following three levels. The first concerns objective characteristics like farmers’ livelihood capital. Research showed that ecological protection behavior of farmland is significantly promoted by the farmer’s individual features and training participation experience [23,24]. The second involves subjective factors such as emotion, attitude, and cognition. Research has shown that environmental knowledge and self-efficacy significantly promoted farmer’s ecological protection of farmland [25,26]. In addition, non-cognitive abilities such as optimism, resilience, and life satisfaction also significantly affect farmland ecological protection [27]. Meanwhile, with regard to the main body of current agricultural production and operation, the aging labor force has insufficient cognitive ability and low ability to learn actively, which is not conducive to transforming green output [28]. The third level involves the external situational characteristics. The academic community generally believes that social norms and the institutional environment are essential factors affecting a farmer’s farmland ecological protection behavior [29,30].

The existing literature has carried out rich research about the impact of human capital on a farmer’s farmland ecological protection. However, there is still room for expansion: First, the impact of human capital on farmer’s farmland ecological protection in the existing literature mainly focuses on explicit human capital, such as education, training, planting years, cognitive ability, etc. There is a lack of analysis of the impact mechanism of NCA on a farmer’s ecological protection of farmland (hereafter, EPF), and its measurement indicators also lack economic meaning. Second, most literature tried to find ways to improve human capital for the aging labor force from external aspects, such as social networks, but ignores the promotion of the human capital of the elderly labor force from the implicit human capital of NCA. Therefore, based on the new human capital theoretical framework, this study uses the data from 964 farmer household surveys, systematically measures the non-cognitive ability according to the “big five” personality traits, and empirically analyzes the influence and mechanism of non-cognitive ability on farmers’ ecological protection of farmland to reveal the vital role of NCA on farmer’s EPF.

## 2. Theoretical Analysis

### 2.1. The Effect of NCA on Farmer’s EPF

Non-cognitive ability (NCA) refers to the relatively stable feeling, thought, and behavior patterns that do not directly affect cognition but play a role in the cognitive process. Both NCA and cognitive abilities (CA) can be improved through education and external interventions [31]. Nevertheless, compared to cognitive abilities, NCA is more sensitive and malleable [32]. Richards divided NCA into three parts: attitude, inner behavior, and social skills [33]. Among them, dimensions such as self-efficacy, social interaction ability, resilience, life satisfaction, and active learning ability have become essential components of NCA [34,35,36]. Therefore, this study explains the influential mechanism of NCA on farmer’s EPF from nine dimensions. Self-efficacy refers to farmer’s subjective judgment and cognition on capital, time, and other dimensions of adopting ecological protection of farmland technology [26]. High-level active learning capabilities can facilitate technology adoption by leveraging rich information channels, reducing risk and uncertainty, and improving crop skills and agricultural income [37]. Stronger social communication abilities can enable farmers to master more practical technical information, reduce the risk of technology adoption, and improve the efficiency of technology adoption [38]. In agricultural production, farmers will consider profit maximization and incorporate risk minimization into their decision-making goals [39]. EPF is a long-term and complex activity, and farmers need high individual resilience and stress resistance to deal with potential risks effectively. Farmers with a high level of resilience can recover soon even if they are frustrated in adopting ecological protection of farmland and ultimately stimulate the economic benefits of technology adoption [40]. An optimistic attitude can stimulate the potential of individuals and enhance their ability to take risks. The more confident farmers are in their future life, the stronger their resilience and ability to resist setbacks when facing various livelihood risks. Farmers can still have confidence, even if the EPF fails to pay off in the short-term [41]. In addition, farmers with higher life satisfaction tend to pursue a higher life level, such as ecological product safety and a better living environment, and the possibility of adopting alternative technologies for ecological protection of farmland will also increase accordingly. EPF has a significant externality. The higher the degree of altruism of farmers, the more concerned they are about others and social welfare, the stronger the motivation to protect the environment [35]. Based on the above analysis, the following assumptions are made.
**Hypothesis** **1** **(H1).***NCA can promote farmer’s EPF.*

### 2.2. Influence Mechanism of NCA on Farmer’s EPF

NCA can directly affect farmer’s EPF and affect EPF through social capital, risk perception, and information channels. Personal behavior decision-making is a relatively complex psychological process that includes multiple factors such as risk perception, attitude, and opportunity recognition. NCA has a crucial impact on the above factors [42]. First, NCA strengthens farmer’s social capital by expanding interpersonal networks and affecting a farmer’s EPF. Individuals with strong NCA are more aggressive, emotionally stable, and have stronger motivation and ability to acquire the knowledge, networks, and asset resources needed for EPF [43,44]. The higher the NCA of farmers, the stronger the interpersonal skills and the more comprehensive the relationship network, which helps transform the relationship advantages into resource advantages and use the better resources in the network to win higher quality external support for technology adoption. Second, NCA can influence EPF by strengthening value perception. The uncertainty of ecological conservation technology is the key to whether farmers adopt it or not. Farmers’ behaviors of adopting EPF faces multi-dimensional risks such as market and production reduction risks, cost input, and so on. NCA can affect farmers’ risk preference and risk perception to a certain extent [45]. Higher NCA can prompt farmers to form correct value perceptions of new technology. The lower NCA of farmers, the less determined they are, the easier it is to exaggerate technical risks, and they will retreat despite difficulties, which is not conducive to the adoption of ecological conservation technology. Finally, NCA can also influence farmer’s EPF through information channels. Studies have shown that strong NCA can help individuals quickly sort out and obtain helpful information from complex information, avoiding depression and irritability [46]. At the same time, farmers’ constraints on information acquisition and capital accumulation are reduced, and credit support is enhanced, thereby improving the ability to pay for the adoption of ecological conservation technology [7]. Based on the above analysis, the following assumptions are made.
**Hypothesis** **2** **(H2).***NCA can promote**EPF**through information channels, social capital**,**and value perception**s*.

## 3. Materials and Methods

### 3.1. Data Sources

The data in this study comes from field surveys conducted by members of the research group in Shaanxi, Sichuan, and Anhui provinces in China from July to August 2020. Shaanxi and Sichuan belong to the backward regions in the west, and Anhui belongs to the more developed regions in the middle and east in China. The three provinces are important tea production areas, and there are significant differences in tea business conditions such as tea brandy, tea scale, and output value, so the study area has good representativeness. The research was conducted using stratified and random sampling methods. First, three to four counties (districts) were selected in each of the three provinces according to regional economic conditions and tea production. Secondly, the township selected two to three administrative villages, and 10–15 farmers were randomly selected in each village for one-on-one interviews. The main contents of the survey questionnaire include NCA, individual and family characteristics of farmers, characteristics of EPF, and external environment characteristics. To improve the validity and authenticity of the data, we selected farmers who are mainly engaged in/familiar with tea as the research objects. A total of 1020 questionnaires were distributed in the survey, 964 valid samples were finally obtained, and the effective rate of the questionnaires reached 94.50%.

### 3.2. Variable Selection and Descriptive Statistics

#### 3.2.1. EPF

According to the specific measures in the Chemical Fertilizer Reduction and Substitution Action in China, combined with the actual production of tea, the technology of chemical fertilizer reduction (CR) and organic fertilizer (OF) was selected to measure the farmer’s ecological protection behavior of farmland. In 2015, the Ministry of Agriculture in China proposed the reduction of chemical fertilizers, so the indicators were measured by asking “whether organic fertilizers are used” and “whether chemical fertilizers have been reduced in the past five years”. The average value of chemical fertilizer reduction is 0.192, and the average value of organic fertilizer application is 0.753 (Table 1), which preliminarily shows that with the implementation of the replacement policy of chemical fertilizer reduction, the proportion of farmers using organic fertilizer has increased dramatically. However, the proportion of chemical fertilizer reduction applications is still low.

#### 3.2.2. NCA

Non-cognitive abilities (NCA) are broader in content, more flexible in their measurement indicators, and have no unified structure. This study used the “Big Five” personality traits to measure NCA [47]. The “Big Five” is the most commonly used and widely accepted measure of non-cognitive abilities, and it can eliminate specific measurement errors [9]. The NCA specifically includes the following variables: self-efficacy, social interaction ability, optimism, resilience, active learning ability, altruistic tendency, and life satisfaction [33,34,35,36]. At the same time, the factor analysis method was used to measure the NCA of the overall dimension (Cronbach’s α was more significant at 0.7, and the KMO value was 0.666). Extracting four common factors whose eigenvalue is greater than 1, the cumulative variance contribution rate reaches 65.361%, indicating that the extracted common factors can better reflect non-cognitive ability.

#### 3.2.3. Mediating Variables and Control Variables

Referring to the research of Wang [13], social capital, information channels, and value perception are selected as mediating variables (MV) in this study. This document is used to select the mediating variable in this study because it has certain similarities and references in terms of variable measurement methods, research neighborhoods, and influence mechanisms. First of all, this study is also based on the “Big Five “personality measures of non-cognitive ability. In addition, this study explores the influence of non-cognitive ability on micro-individual behavior and decision-making results like this study, and has a certain similarity in the impact mechanism. Therefore, this study considers this literature as the primary reference when selecting mediating variables. Moreover, this study selected 13 variables (AH, EL, etc.) as control variables (CV) from the aspects of individual characteristics, family characteristics, environment characteristics [48,49] (Table 1).

### 3.3. Methods

#### 3.3.1. Benchmark Model

In this study, EPF includes the two behaviors of chemical fertilizer reduction (CR) and organic fertilizer application (OF). Both of these two types of decisions are discrete binary variables. However, they may be affected by many similar factors when making specific decisions, that is, the two behaviors are not independent, and the combined action of the two can produce the following results: Neither chemical fertilizers nor organic fertilizers are applied, only chemical fertilizers are reduced, only organic fertilizers are applied, and both chemical fertilizers and organic fertilizers are applied. If $y_{1}$ is used to represent the decision of the farmer to reduce the application of chemical fertilizer, then $y_{1}=1$ and $y_{1}$ = 0 indicates that the farmer reduces the application of chemical fertilizer and does not reduce the application of chemical fertilizer, respectively. Similarly, $y_{2}$ is used to represent the decision to apply organic fertilizer, then $y_{2}=1$ and $y_{2}=0$ indicate the farmer applies organic fertilizer and does not apply organic fertilizer, respectively, then the above four results can be expressed as (0, 0), (0, 1), (1, 0), (1, 1). The bivariate probit model can allow correlation between the error terms of differential equations. Therefore, this study analyzes the factors that affect a farmer’s reduction in CR and OF by constructing a bivariate Probit model. The specific form is as follows:(1)$$\{\begin{array}{l} y_{1i}^{*}=\gamma_{1}^{\prime }X_{i}+\varepsilon_{1i}\\ y_{2i}^{*}=\gamma_{2}^{\prime }X_{i}+\varepsilon_{2i} \end{array}$$

Among them, $y_{1i}^{*}$ and $y_{2i}^{*}$ represent the choice of farmers to reduce chemical fertilizer application and organic fertilizer application respectively, *i* = 1, 2, *n* represents the *i*-th sample; $X_{i}$ represents the respective variables that affect farmer’s CR and OF; $\gamma_{1}^{\prime }$ and $\gamma_{2}^{\prime }$ represents the corresponding estimated coefficients.

For latent variables $y_{v}^{*}$, suppose:(2)$$y_{j}=\{\begin{array}{l} 1,y_{v}>0\\ 0,\mathrm{others} \end{array}$$

If the farmer’s decision of CR and OF is independent, the above two formulas are univariate Probit models, and $\varepsilon_{v1}$ and $\varepsilon_{v2}$ are independent and identically distributed; but if farmer make decision of CR and OF at the same time, and the two decisions are not mutually exclusive, $\varepsilon_{v1}$ and $\varepsilon_{v2}$ obey the normal distribution, and the corresponding covariance matrix ψ is:(3)$$\psi =\left[\begin{array}{cc} 1 & \rho_{12}\\ \rho_{21} & 1 \end{array}\right]$$

In Formula (3), if the value on the off-diagonal line is not equal to 0, it indicates a correlation between the farmer’s decision to reduce chemical fertilizer application and organic fertilizer application, and the bivariate probit model is suitable for this case.

#### 3.3.2. The Mediation Effect Model

This study constructs the following mediation effect model [50,51]:(4)$$GT_{i}=\alpha_{0}+\alpha_{1}X_{i}+\sum \alpha_{2}Z_{i}+\varepsilon_{1}$$
(5)$$EE_{i}=\beta_{0}+\beta_{1}X_{i}+\sum \beta_{2}Z_{i}+\varepsilon_{2}$$
(6)$$GT_{i}=\gamma_{0}+\gamma_{1}X_{i}+\gamma_{2}EE_{i}+\sum \gamma_{3}Z_{i}+\varepsilon_{3}$$

In formulas (4)–(6), represents EPF, $EE_{i}$ represents MV, $X_{i}$ represents the core variable NCA; $Z_{i}$ is CV; *α*, *β* and *γ* are parameters to be estimated, respectively, and $\varepsilon_{1}$, $\varepsilon_{2}$, $\varepsilon_{3}$ are residual items.

## 4. Results

Using Stata 14.0 software (StataCorp LP, 4905 Lakeway Drive, College Station, TA, USA), based on the survey data of tea farmers in the Shaanxi, Sichuan and Anhui provinces in China, the bivariate probit model was used to test the influence of NCA on farmer’s EPF (Table 2). Model 1 only includes control variables, model 2 introduces the NCA of the overall dimension based on the CV, and model 3 introduces the variables of each dimension of NCA based on model 1. The *p*-value has passed at least 10% of the significance test. The likelihood-ratio has also passed at least 5% of the significance test, indicating that the decision of farmers to reduce chemical fertilizer and use an organic fertilizer is related. The value of *p* is positive, indicating that CR and OF have complementary effects, that is, the probability of farmers who reduce the application of chemical fertilizers to apply organic fertilizer at the same time is greater than that of farmers who do not reduce the application of chemical fertilizers.

### 4.1. The Effect of NCA on Farmer’s EPF

The NCA of the overall dimension promotes farmer’s CR and OR at the 1% significance level (Table 2), indicating that NCA is an important factor that significantly promotes a farmer’s farmland ecological protection behavior. From the perspective of different dimensions, the interaction with village cadres and the government significantly promoted the reduction of CR and OF. The strong social communication ability can promote farmers to obtain richer knowledge, network and asset resources through the government and village cadres, which is beneficial to farmers in making decisions on the adoption of EPF. Active learning ability (mean: 3.184) significantly promotes EPF at the 5% statistical level, which illustrates that the farmer’s learning initiative helps farmers to master the technology as soon as possible and improve its probability of adoption. Self-efficacy significantly promotes chemical fertilizer reduction behavior, but it is not significant for organic fertilizer application. The stronger the self-efficacy, the higher the farmer’s perception of the knowledge and ability to master the ecological protection technologies. Due to the separation of farming policies, it becomes more challenging to obtain organic fertilizers (especially in mountainous areas), thus inhibiting the positive effect of self-efficacy to a certain extent. The ability to resist stress significantly promotes the reduction of chemical fertilizer application because the adoption of new technologies faces huge uncertainty, and the strong ability to withstand pressure can enable farmers to enrich the choice of risk defense strategies, thereby increasing the probability of EPF. The altruistic tendency significantly promotes fertilizer reduction behavior. Generally, farmers with higher altruistic tendencies pay more attention to others and to social welfare, and their motivation to protect the environment will also increase accordingly. Individual resilience significantly promotes the chemical fertilizer reduction behavior of farmers. This is because the higher the resilience index of farmers, the stronger the ability to quickly recover from the production and operation difficulties, so they tend to adopt ecological protection of farmland technology. Thus, Hypothesis 1 is partially verified.

The older the household is, the more serious the land plot, and the more inclined a farmer is to reduce the application of chemical fertilizers to maintain soil fertility. Green subsidies significantly promote the reduction of fertilizer application, indicating that government subsidies reduce a farmer’s input costs for green production technology to a certain extent, which increases the enthusiasm for reducing fertilizer application. Cognitive ability significantly promoted OF, indicating that the stronger the cognitive ability, the greater the probability of applying organic fertilizer. The agricultural labor force significantly promoted OF, indicating that more labor can help farmers improve the probability of farmland protection. The agricultural labor force hurts CR. A farmer’s degree of concurrent operation is generally high. In order to minimize the loss of opportunity costs of farming, it is not conducive to the reduction of fertilizer application. The tea planting years and the neighborhood effect significantly promoted CR and OF. The longer planting years (mean 24.387 years) and more neighborhood references help improve farmland ecological protection. The distance to the county town significantly promotes CR. With the construction of urban and rural transportation, it is more and more convenient for farmers to enter the county town, which broadens their horizons and helps them accept new ideas and new technologies.

### 4.2. Influence Mechanism Analysis

In this study, a Soble test was used to analyze the influence mechanism of NCA on farmer’s CR through social capital, information channels, and value perception. The mediating effects of social capital, information channels, and value perception passed the significance test of 5%, 1%, and 1%, respectively, in the influence of NCA on EPF (Table 3). Non-cognitive ability strengthens a farmer’s EPF through expanding interpersonal networks, increasing information channels, and improving farmer’s perception of the value of technology adoption. The mediating effect from high to low is information channel (19.27%), value perception (11.09%), and social capital (5.27%).

It can be seen from Table 4 in the process of non-cognitive ability (NCA) influencing OF that the mediating effects of social capital, information channels, and value perception passed the significance test of 5%, 1%, and 5%, respectively, which also shows that the mediating effects of capital, information channels and value perception are significant. Thus, H2 is verified.

### 4.3. Heterogeneity Analysis

At present, the middle-aged and elderly labor force is the leading force in Chinese agricultural production. In the sample of this study, peasant households aged 50 and below only account for 25.52% of the total sample, and those aged between 50 and 70 account for as high as 64.11% of it, indicating that farmers aged 50–70 are the main body of the sample. Farmers of different ages and income levels may have significant differences in NCA due to differences in educational level, values, and employment opportunities. Therefore, this study divides the samples into two variables, total household income and the number of agricultural laborers over 60 years old (including), to explore the income heterogeneity and age heterogeneity of the influence of NCA on farmer’s EPF. Interaction with the government, active learning ability, self-efficacy, altruistic tendency, and optimism in the low-income group significantly promote the behavior of farmer’s fertilizer reduction (Table 5). In contrast, only the high-income group can resist stress. This shows that NCA has a greater impact on the low-income group. Compared with low-income farmers, farmers with higher incomes face little living pressure and uncertain income. At the same time, they have a more substantial economic level, resource accessibility, risk-taking ability, and sufficient ability to purchase green production materials. Therefore, NCA has a more significant impact on a farmer’s ecological protection behavior of farmland. In addition, the NCA of the elderly group has a greater impact on farmer’s ecological protection behavior of farmland in general. It is difficult for older workers to improve their human capital through channels like training, and NCA can help increase technology adoption by improving life satisfaction among older adults.

## 5. Discussion

Based on the perspective of new human capital, this study incorporates NCA into the analysis framework of a farmer’s EPF. Through empirical analysis, it is found that NCA has a significant role in promoting EPF. Among them, social communication ability, active learning ability, self-efficacy, stress resistance, altruistic tendency, and resilience significantly promoted farmer’s EPF, which was consistent with the research conclusions of Rauch [18] and Chen [27]. In addition, research shows that NCA can also promote EPF by broadening information channels, enhancing a farmer’s social capital, and referring to the level of farmers’ perceived value of technology adoption. This shows that NCA is an essential factor affecting EPF. China is currently in a critical period of social transformation; the traditional rigid governance model makes farmer’s green production behavior passive and generally has little effect. NCA stimulates farmer’s enthusiasm from the perspectives of resilience, stress resistance, active learning, and social interaction, helping to provide a lasting impetus for promoting the green transformation of agriculture.

This study further shows that NCA has a greater impact on the EPF of farmers in the low-income group and the elderly group, which is consistent with Li and Chen [52]. The high degree of aging of the agricultural labor force and the difficulty in increasing the income of small farmers are the primary national conditions of China. At present, the middle-aged and elderly labor force has become a key force in the green transformation of agriculture. However, their explicit human capital, such as cognitive ability, is showing a decline [28]. Therefore, it is difficult for the elderly labor force to improve their human capital through training and other channels. As a kind of implicit human capital, NCA has a more prominent influence on the behavioral decision-making of individuals with relatively low cognitive ability [53,54]. NCA can improve the life satisfaction of the elderly labor force, relieve depression, social adaptability, and increase their level of adoption of EPF. This study verifies that NCA has a more significant impact on the aging and low-income farmer’s EPF, which promotes the adoption of farmland ecological protection technologies for aging and low-income farmers.

The possible shortcomings of this study are: First, in this study, NCA is the core explanatory variable. Although cognitive ability is included in the model as a control variable, it does not explore the relationship between NCA and cognitive ability due to space limitations, which is a direction that the authors will study later. Second, due to the data availability and other factors, tea as a cash crop is taken as an example, but other economic crops and food crops are not considered. Third, farmland ecological protection technology includes a variety of technologies. Considering the current farmland ecological protection technology promotion in China in recent years and the actual production of tea in local areas, we only selected chemical fertilizer reduction technology and organic fertilizer technology to represent the farmland ecological protection technology.

## 6. Conclusions

Based on the data of 964 tea growers, this study empirically tested the effect of NCA on EPF by systematically measuring NCA and paying attention to the role of social capital, information channels, and value perception. The research shows that, first, NCA is an important factor that significantly promotes EPF. Social communication ability, active learning ability, self-efficacy, stress resistance, altruistic tendency, and resilience significantly were found to promote a farmer’s fertilizer reduction behavior; and social communication ability and active learning ability significantly promoted a farmer’s organic fertilizer application behavior. Therefore, farmers can improve their ecological protection level through social communication ability, active learning ability, social responsibility, and individual resilience. Second, NCA promotes farmer’s EPF by expanding social capital, increasing information channels, and improving the value perception level of technology adoption. Moreover, the mediating effect is information channels, value perception, and social capital from high to low. Third, there is heterogeneity in the influence of NCA on the EPF of farmers, and NCA has a more significant impact on the EPF of farmers in the low-income group and the elderly group. The above conclusions are helpful to provide countermeasures and suggestions for improving farmland ecological protection under the background of low farmer income and aging.

Based on the above conclusions, the following implications can be made: First, considering the role of traditional human capital such as farmer’s cognitive ability, planting years, and labor force, the improvement of a farmer’s NCA should be given full attention. The room for farmer’s cognitive ability improvement is limited, while NCA still has excellent plasticity. Therefore, the government should include NCA training in the existing training system to strengthen the practical guidance of a farmer’s self-confidence, active learning ability, social interaction ability, and other personality characteristics to improve a farmer’s non-cognitive ability. Second, it is very necessary to focus on the critical role of social capital, information channels, and value perception in the process of NCA affecting EPF. It is essential to link rural grassroots organizations, actively encourage and organize farmers to participate in collective activities, and strengthen exchanges between farmers. Furthermore, guiding farmers to use mobile phones, the internet, and other means enhances the heterogeneity of agricultural information absorption and expands information channels. In addition, increasing the publicity and training of a farmer’s ecological production technology improves the farmer’s value perception level of technology. We must pay attention to the heterogeneity of the influence of NCA on EPF. For farmers in the low-income group and the elderly group, especially the elderly labor force, the guidance of NCA should be strengthened to enhance the life satisfaction of farmers to induce the EPF.

## Figures and Tables

**Table 1 ijerph-19-07598-t001:** Variable description and descriptive statistics.

Variable	Item	Content	Mean	Standard Deviation
EPF	CR	Yes = 1, no = 0	0.192	0.394
	OF	Yes = 1, no = 0	0.753	0.431
NCA	LS	Satisfaction with your life ^a^	3.382	0.669
	IV	Do you frequently interact with village cadres and cooperatives ^a^	2.568	1.088
	IG	Do you frequently interact with government officials ^a^	2.041	1.014
	IA	Actively learn the new technical knowledge ^a^	3.184	1.034
	SE	Master the information channel of ecological protection technology of farmland ^a^	2.839	0.861
	CA	Your family’s income in the village ^b^	2.950	0.724
	AT	The adoption of ecological protection technology of farmland has great social benefits, and I am very proud of it ^a^	3.806	0.787
	OI	As long as you work hard, life will get better and better ^a^	3.240	0.814
	TI	Believe that the way is better than the difficulty ^a^	3.671	0.705
MV	SC	The number of households that are usually close to each other/Household	24.276	29.685
	IC	Number of channels for obtaining tea prices/individual	2.405	0.757
	VP	Compared with the price benefit of tea, it is worthwhile to pay for the replacement technology of farmland ecological protection ^a^	3.016	0.934
CV	AH	age	57.381	10.276
	EL	Education level of the head of the household. Elementary school and below = 1; junior high school = 2; high school = 3; high school or above = 4	1.408	0.620
	AA	Is scientific fertilization important ^a^	3.941	0.530
	TP	Whether to participate in green production training. Yes = 1, no = 0	0.524	0.499
	TA	year	24.387	14.991
	AL	people	1.942	0.721
	BS	mu	6.176	7.802
	TR	million	9.702	13.153
	GS	Green fertilizers such as organic fertilizers can be distributed in time ^a^	2.821	0.975
	NE	Do neighbors adopt ecological protection of farmland technology ^c^.	2.928	0.995
	DC	km	33.574	18.905
	SX	Yes = 1, no = 0	0.394	0.488
	SC	Yes = 1, no = 0	0.306	0.461

^a,b,c^ all represent a five-point Likert scale. ^a^: strongly disagree = 1 to strongly agree = 5; ^b^: lower layer = 1, lower middle layer = 2, middle layer = 3, upper middle layer = 4, upper layer = 5; ^c^: none = 1, some adopt = 2, few adopt = 3, majority adopt = 4, all adopt = 5.

**Table 2 ijerph-19-07598-t002:** Estimated results of the effect of NCA on farmer’s EPF.

Variable	Model 1		Model 2		Model 3	
	CR	OF	CR	OF	CR	OF
NCA			0.795 *** (0.119)	0.568 *** (0.107)		
LS					0.032 (0.100)	0.054 (0.083)
IV					0.001 (0.073)	0.121 * (0.063)
IG					0.172 ** (0.075)	0.128 * (0.073)
IA					0.138 ** (0.068)	0.130 ** (0.055)
SE					0.154 ** (0.069)	0.047 (0.064)
CA					0.155* (0.092)	0.019 (0.080)
AT					0.181 ** (0.082)	0.071 (0.067)
OI					0.117 (0.077)	0.055 (0.067)
TI					0.172 * (0.093)	0.028 (0.075)
AH	0.007 (0.005)	−0.004 (0.004)	0.008 (0.005)	−0.004 (0.004)	0.010 * (0.005)	−0.004 (0.004)
EL	0.038 (0.088)	−0.019 (0.080)	0.001 (0.091)	−0.042 (0.081)	−0.009 (0.093)	−0.032 (0.082)
AA	0.239 ** (0.110)	0.207 ** (0.089)	0.155 (0.117)	0.152 * (0.090)	0.160 (0.120)	0.174 * (0.092)
TP	0.005 (0.114)	0.312 ** (0.099)	0.200 (0.123)	0.182* (0.103)	0.225 (0.136)	0.122 (0.111)
AL	−0.094 (0.079)	0.166 ** (0.070)	−0.160 (0.082)	0.159 ** (0.070)	−0.144* (0.084)	0.164 ** (0.071)
TP	0.234 ** (0.084)	0.012 (0.064)	0.230 ** (0.086)	0.017 (0.065)	0.218 ** (0.087)	0.014 (0.065)
BS	−0.013 (0.075)	−0.036 (0.067)	−0.065 (0.078)	−0.084 (0.069)	−0.075 (0.079)	−0.083 (0.070)
TR	0.002 (0.004)	0.010 ** (0.004)	−0.001 (0.004)	0.008 * (0.004)	−0.002 (0.004)	0.008 * (0.004)
GB	0.262 *** (0.060)	0.142 ** (0.052)	0.177 ** (0.064)	0.088 (0.054)	0.176 ** (0.065)	0.094 * (0.055)
NE	0.349 *** (0.064)	0.466 *** (0.053)	0.281 *** (0.067)	0.416 *** (0.055)	0.272 *** (0.070)	0.443 *** (0.057)
DC	0.431 *** (0.094)	0.067 (0.087)	0.404 *** (0.098)	0.047 (0.088)	0.421 *** (0.099)	0.065 (0.090)
SX	−0.181 (0.142)	−0.158 (0.143)	−0.317 ** (0.150)	−0.193 (0.146)	−0.250 (0.157)	−0.238 (0.154)
SC	−1.214 *** (0.184)	0.215 (0.149)	−1.232 *** (0.189)	0.228 (0.150)	−1.159 *** (0.200)	0.207 (0.158)
Constant term	−5.883 *** (0.873)	−2.304 *** (0.712)	−4.750 *** (0.923)	−1.536 ** (0.733)	−8.314 *** (1.070)	−2.954 *** (0.828)
N	964		964		964	
*p*	0.241 * (0.083)		0.143 * (0.086)		0.173 * (0.088)	
Likelihood-ratio test	8.591 **		2.770 *		3.875 **	
Wald chi2	275.78 ***		324.89 ***		333.80 ***	
Log-likelihood	−822.900		−786.565		−778.388	

***, **, * indicate significance at the 1%, 5% and 10% levels, respectively; standard errors are in brackets.

**Table 3 ijerph-19-07598-t003:** Influence mechanism test of NCA on CR.

Variable	CR	SC	CR	IC	CR	VP	CR
NCA	0.221 *** (0.022)	0.249 *** (0.046)	0.210 *** (0.023)	0.605 *** (0.041)	0.179 *** (0.025)	0.484 *** (0.053)	0.197 *** (0.023)
MV			0.046 ** (0.015)		0.070 *** (0.017)		0.050 *** (0.013)
CV	YES	YES	YES	YES	YES	YES	YES
Constant term	0.186 *** (0.026)	2.523 *** (0.055)	0.067 (0.047)	2.449 *** (0.048)	0.013 (0.050)	2.755 *** (0.062)	0.046 (0.046)
MV’s significance	0.011 ** (0.004)			0.042 *** (0.011)		0.024 *** (0.007)	
Mediation effect of proportion/%	5.27			19.27		11.09	

The intermediary variables correspond to SC, IC and VP respectively; ***, ** indicate significance at the 1% and 5% levels, respectively; standard errors are in brackets.

**Table 4 ijerph-19-07598-t004:** Influence mechanism test of NCA on OF.

Variable	OF	SC	OF	IC	OF	VP	OF
NCA	0.208 *** (0.025)	0.249 *** (0.046)	0.198 *** (0.025)	0.605 *** (0.041)	0.167 *** (0.027)	0.484 *** (0.053)	0.188 *** (0.026)
MV			0.038 ** (0.017)		0.070 *** (0.019)		0.041 ** (0.015)
CV	YES	YES	YES	YES	YES	YES	YES
Constant term	0.743 *** (0.029)	2.523 *** (0.055)	0.645 *** (0.052)	2.449 *** (0.048)	0.570 *** (0.056)	2.755 *** (0.062)	0.629 *** (0.051)
MV’s significance	0.009 ** (0.004)			0.042 *** (0.012)		0.019 ** (0.007)	
Mediation effect of proportion/%	4.62			20.48		9.59	

***, ** indicate significance at the 1% and 5% levels, respectively; standard errors are in brackets.

**Table 5 ijerph-19-07598-t005:** Heterogeneity analysis.

Variable	CR		OF		CR		OF	
	Low Income	High Income	Low Income	High Income	Young Age	Old Age	Young Age	Old Age
LS	−0.030	0.274	0.058	0.037	0.003	0.028	−0.049	0.187
	(0.123)	(0.208)	(0.096)	(0.200)	(0.140)	(0.152)	(0.118)	(0.121)
IV	0.011	0.019	0.153 **	0.120	0.029	0.061	0.079	0.209 **
	(0.096)	(0.130)	(0.077)	(0.131)	(0.102)	(0.111)	(0.087)	(0.098)
IG	0.197 **	0.122	0.120	0.150	0.168 *	0.214 *	0.210 **	0.022
	(0.100)	(0.131)	(0.087)	(0.151)	(0.101)	(0.122)	(0.099)	(0.112)
IA	0.264 **	0.015	0.160 **	0.103	0.096	0.237 **	0.041	0.256 **
	(0.091)	(0.113)	(0.067)	(0.110)	(0.093)	(0.110)	(0.074)	(0.088)
SE	0.245 **	0.013	0.040	0.342 **	0.217 **	0.091	0.150 *	−0.078
	(0.092)	(0.115)	(0.078)	(0.128)	(0.098)	(0.103)	(0.090)	(0.094)
CA	0.072	0.308 *	−0.078	0.192	0.114	0.165	0.122	0.188
	(0.121)	(0.164)	(0.096)	(0.172)	(0.124)	(0.151)	(0.110)	(0.125)
AT	0.407 ***	0.049	−0.131	0.069	0.176	0.187	0.024	−0.163
	(0.114)	(0.134)	(0.080)	(0.142)	(0.118)	(0.121)	(0.097)	(0.098)
OI	0.205 *	0.029	0.021	0.183	0.032	0.232 *	0.115	−0.021
	(0.106)	(0.129)	(0.081)	(0.136)	(0.109)	(0.120)	(0.094)	(0.101)
TI	0.142	0.186	0.112	−0.132	0.121	0.200	0.040	0.045
	(0.120)	(0.164)	(0.089)	(0.158)	(0.133)	(0.142)	(0.106)	(0.113)
CV	YES	YES	YES	YES	YES	YES	YES	YES
Constant term	−9.957 ***	−7.402 ***	−3.510 ***	−2.519	−8.997 ***	−6.323 ***	−3.126 **	−3.425 **
	(1.537)	(1.844)	(0.986)	(1.970)	(1.408)	(1.395)	(1.009)	(1.082)
N	633	331	633	331	528	436	528	436
LR chi2	212.35	87.22	126.28	96.69	164.65	122.39	104.79	105.30
Pseudo R2	0.354	0.253	0.169	0.300	0.307	0.298	0.184	0.207
Log-likelihood	−193.818	−128.465	−310.066	−112.755	−185.178	−143.600	−231.807	−200.932

***, **, * indicate significance at the 1%, 5% and 10% levels, respectively; standard errors are in brackets.

## Data Availability

The data used in this study is are available on request from the corresponding author.

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
