# Peer review of "The Effect of Non-Cognitive Ability on Farmer’s Ecological Protection of Farmland: Evidence from Major Tea Producing Areas in China"

_ijerph, 2022, doi:10.3390/ijerph19137598_

Round 1
Reviewer 1 Report
- In the Abstract section, the sentences in lines 1-2 appear to be misplaced from the left margin.
- Similarly, in the Introduction section, the sentences in lines 2-7 are also misplaced from the margin of the paragraph.
- Page 2, paragraph 2, and line 16: please edit.
- Page 4 line 10: add a space before the sentence. Same page line 37: add a space before the sentence.
- Page 13 (Conclusion section), paragraph 2 line 9: edit the text.
- Page 14, line 2: edit.
Overall, this is high quality manuscript, which has scientific as well as practical implications.

Author Response
Response to Reviewer 1 Comments
Point 1: English language and style are fine/minor spell check required.
Response 1: I am very grateful to the reviewer 1 for your affirmation of the language of this article. The author has re-checked the original text carefully and corrected the spelling errors.
Point 2: Comments and Suggestions for Authors:
Response 2: Many thanks to the reviewer 1 for your very valuable comments on the revision of this article.
Point 3: In the Abstract section, the sentences in lines 1-2 appear to be misplaced from the left margin.
Response 3: The author uses the latest manuscript provided by the editorial office as a revision, in this version the left margins of the abstract and introduction have been readjusted. Thanks for pointing out the errors in the paper.
Point 4: Similarly, in the Introduction section, the sentences in lines 2-7 are also misplaced from the margin of the paragraph.
Response 4: The author uses the latest manuscript provided by the editorial office as a revision, in this version the left margins of the abstract and introduction have been readjusted.
Point 5: Page 2, paragraph 2, and line 16: please edit.
Response 5: The author has edited line 16 of paragraph 2 on page 2.
Point 6: Page 4 line 10: add a space before the sentence. Same page line 37: add a space before the sentence.
Response 6: I'm so sorry, I can't seem to find where the sentences on page 4, line 10 and line 37 should have spaces. Probably it is because the revision I'm currently using was handled by the editor. If it's my own reason that I didn't find, please ask the expert to let me know in the follow-up. I'm sorry for the trouble it may have caused.
Point 7: Page 13 (Conclusion section), paragraph 2 line 9: edit the text.
Response 7: The author has edited line 9 of paragraph 2 on page 13 (conclusion).
Point 8: Page 14, line 2: edit.
Response 8: The author has edited line 2 on page 14.
Point 9: Overall, this is high quality manuscript, which has scientific as well as practical implications.
Response 9: Finally, I would like to thank the review 1 again for your recognition of the significance of this paper and the above valuable comments.
Reviewer 2 Report
The current study is on a topic of general interest to the readers of International Journal of Environmental Research and Public Health. I found the paper to be overall well written and much of it to be well described. Below I have some comments for the Authors to address:
I am not truly convinced that the outcome of the study is the index that could be further validate and used in other studies. The study gives a broad and interesting picture of Chinese farmers’ non cognitive skills and their relevance in ecological protection of farmland but it does not go beyond that. I would recommend to use other terminology than index.
Title:
I would recommend reconsidering the title “The Effect of Non-Cognitive Ability on Farmer's Ecological Protection of Farmland: Evidence from Chinese Rural” e.g. by adding the region where the study was performed. As such it seems incompleted. Rural what?
Material and methods
I miss more information on it how the items included in the measurement instrument were developed and pretested.
Please provide the questionnaire used in the study as a supplementary material.
The EPF is measured with reference to the chemical fertilizer reduction (CR) and organic fertilizer (OF) use. It should be considered as the limitation of the study and any recommendation on other variables to consider should be added as concluding remarks.
Result
Add more details on the profile of the sample
Author Response
Response to Reviewer 2 Comments
Point 1: Comments and Suggestions for Authors:
Response 1: Many thanks to your very valuable comments on this article.
Point 2: The current study is on a topic of general interest to the readers of International Journal of Environmental Research and Public Health. I found the paper to be overall well written and much of it to be well described. Below I have some comments for the Authors to address:
Response 2: I would like to thank the reviewer 2 for your recognition of the research value of this paper and for their very helpful revisions.
Point 3: I am not truly convinced that the outcome of the study is the index that could be further validate and used in other studies. The study gives a broad and interesting picture of Chinese farmers’ non cognitive skills and their relevance in ecological protection of farmland but it does not go beyond that. I would recommend to use other terminology than index.
Response 3: Comparing with the opinions of the reviewers, the original statement "building a non-cognitive ability indicator system" may not be rigorous enough. Therefore, drawing on the opinions of reviewers, this paper replaces the original statement with "systematic measurement of non-cognitive abilities of core variables", and makes corrections in the introduction and conclusion of the text.
Point 4: [Title] I would recommend reconsidering the title “The Effect of Non-Cognitive Ability on Farmer's Ecological Protection of Farmland: Evidence from Chinese Rural” e.g. by adding the region where the study was performed. As such it seems incompleted. Rural what?
Response 4: Acknowledging the opinions of the reviewers, the original title was revised to“The Effect of Non-Cognitive Ability on Farmer's Ecological Protection of Farmland: Evidence from Major Tea Producing Areas in China”.
Point 5: [Material and methods] I miss more information on it how the items included in the measurement instrument were developed and pretested. Please provide the questionnaire used in the study as a supplementary material.
Response 5: The author added annotations to Table 1, supplemented the measurement dimensions of some key variables, and added a questionnaire as supplementary materials (see appendix).
Point 6: The EPF is measured with reference to the chemical fertilizer reduction (CR) and organic fertilizer (OF) use. It should be considered as the limitation of the study and any recommendation on other variables to consider should be added as concluding remarks.
Response 6: Acknowledging the opinions of reviewers. In the Discussion section, the authors further supplemented the limitations of the dependent variable measure in this paper. In addition, added concluding observations on relevant variables in the conclusion section.
Point 7: [Result] Add more details on the profile of the sample.
Response 7: The author further added some detailed descriptions in the result and the sample selection section. For details, see the corresponding section in red font.

Reviewer 3 Report
General:Please re-work introduction, now it is difficult to understand why you choose to study NCA and cognitive, please make link beteween your issues and concepts.
In abstract: which country for farmers?
In abstract: why wo choose to non-cognitive ability? please justify
Also: Specifically, social communication ability, active learning ability, self-efficacy, stress resistance, altruistic tendency and individual resilience signifi-cantly promoted ecological protection of farmland. Please indicate that elements are a part of non cognitive ability.
INTRODUCTION: China has been entering a: please indicate when?
Schultz (p2) please indicate the date
"With the development of cognitive psychology and personality psychology this year," it is not a new domain, please justify this development, where, about what?
"Considering the lessons from the traditional human capital theory that the formation process of human capital is also essential [5], as well as education, skills, and health, the new human capital constructs an individual behavior analysis framework centered on ability, including cognitive and non-cognitive ability (NCA) [6], with particular emphasis on NCA’s prominent role [4]." Cognitive and non cognitive ability and human capital are improperly introduced. Please re-work this paragrapg and the presentation of these concepts in link with your issues.
Cite more references for NCA
"NCA plays a prominent role, especially for low- skilled and middle-skilled groups [10], which profoundly impact the individual’s behavior and decision-making." Please developp, give more examples
"Due to the difference in resilience index, life satisfaction, and self-confidence of different kinds of farmers, there are differences in the decision-making of green production technology adoption of different types of farmers [16]." Why you present resilience index and other. Maybe is it a link with NJCA? Please, can you rework the cohenrence, and links between concepts and that alongside the introduction.
3.2.2
Please justify why you choose to measure NCA with big five
Mediating Variables and Control Variables
Referring to the research of Wang [38],: explain result's of wang and why is it important and does it influence the choice here?

Author Response
Response to Reviewer 3 Comments
Point 1: Comments and Suggestions for Authors:
Response 1: Many thanks to your very valuable comments on this article.
Point 2: General: Please re-work introduction, now it is difficult to understand why you choose to study NCA and cognitive, please make link beteween your issues and concepts.
Response 2: The author has rewritten the introduction. In the second paragraph of the introduction, the author emphasizes the reasons for the study of non-cognitive ability and the relationship between non-cognitive ability and farmland ecological protection based on the practical problems of Chinese agricultural green transition.
Point 3: In abstract: which country for farmers?
Response 3: In the abstract section, the author adds the country name of the study area. Specifically“based on the above background, using the survey data of 964 farmers in China”.
Point 4: In abstract: why wo choose to non-cognitive ability? please justify
Response 4: In the abstract section, the author further adds the reason for choosing non-cognitive ability as the core explanatory variable of this paper (beginning of the second sentence).
Point 5: Also: Specifically, social communication ability, active learning ability, self-efficacy, stress resistance, altruistic tendency and individual resilience signifi-cantly promoted ecological protection of farmland. Please indicate that elements are a part of non cognitive ability.
Response 5: In the findings section of the abstract, the author further noted that variables such as social communication ability, active learning ability, self-efficacy, stress resistance, altruistic tendency and individual resilience are components of non-cognitive abilities. Specifically modified to"Specifically, among the variables of non-cognitive ability, social communication ability, active learning ability, self-efficacy, stress resistance, altruistic tendency and individual resilience significantly promoted ecological protection of farmland”.
Point 6: INTRODUCTION: China has been entering a: please indicate when?
Response 6: Supplement the time of the first sentence of the introduction, specifically" China has been entering a particular period of green production transformation since 2005”.
Point 7: Schultz (p2) please indicate the date
Response 7: Schultz 1964[3] noted the investments in human capital are key to transform traditional agriculture. And further supplemented references.
Point 8: "With the development of cognitive psychology and personality psychology this year," it is not a new domain, please justify this development, where, about what?
Response 8: In the third paragraph of the introduction, the author further supplements and revises the research status of non-cognitive abilities and the rationality of their development.
Point 9: "Considering the lessons from the traditional human capital theory that the formation process of human capital is also essential [5], as well as education, skills, and health, the new human capital constructs an individual behavior analysis framework centered on ability, including cognitive and non-cognitive ability (NCA) [6], with particular emphasis on NCA’s prominent role [4]." Cognitive and non cognitive ability and human capital are improperly introduced. Please re-work this paragrapg and the presentation of these concepts in link with your issues.
Response 9: The author has reorganized the introduction. Among them, the first paragraph is to explain the practical background and research significance of China's agricultural land ecological protection technology. The second paragraph is based on the background of the significant reduction of explicit human capital, such as the cognitive ability of the aging Chinese agricultural labor force, and leads to the important role of non-cognitive ability in the ecological protection of agricultural land of the aging labor force. On the basis of the foregoing, the third paragraph introduces the concept of non-cognitive ability and the research status of non-cognitive ability, and emphasizes its important role in individual behavioral decision-making.
Point 10 Cite more references for NCA
Response 10: The authors have cited more non-cognitive literature, specifically Ref.7,9,13,14 and 15.
Point 11: "NCA plays a prominent role, especially for low- skilled and middle-skilled groups [10], which profoundly impact the individual’s behavior and decision-making." Please developp, give more examples
Response 11: "NCA plays a prominent role, especially for low- skilled and middle-skilled groups [10], which profoundly impact the individual’s behavior and decision-making." The author adds examples to illustrate. The details are as follows: Some studies believe that the influence of NCA on individual behavior even exceeds the role of traditional human capital such as education and cognitive ability, especially for low-and middle-skilled people [18]. For example, research has found that in a labor market characterized by low skills, non-cognitive abilities such as employee loyalty, obedience and persistence are more valuable to employers than cognitive abilities such as intelligence [9, 19]. Wang [20] found that non-cognitive abilities give older people a more significant in digital integration and network access. In addition, NCA can give older workers a greater advantage in life satisfaction, alleviate depression, and improve their social resilience, thereby positivelly impacting farmland ecological protection behavior.
Point 12: "Due to the difference in resilience index, life satisfaction, and self-confidence of different kinds of farmers, there are differences in the decision-making of green production technology adoption of different types of farmers [16]." Why you present resilience index and other. Maybe is it a link with NJCA? Please, can you rework the cohenrence, and links between concepts and that alongside the introduction.
Response 12: The modifications are as follows: In addition, non-cognitive abilities such as optimism, resilience, and life satisfaction also significantly impact farmland ecological protection [26].
Point 13: Please justify why you choose to measure NCA with big five(3.2.2)
Response 13: The study further explains the reasons for using the "Big Five" method to measure non-cognitive abilities.This study used the “Big Five” personality to measure NCA [46].The "Big Five" is the most commonly used and widely accepted measure of non-cognitive abilities, and it can eliminate specific measurement errors [8]. Therefore, non-cognitive ability is measured based on the "Big Five" method.
Point 14: [Mediating Variables and Control Variables] Referring to the research of Wang [38],: explain result's of wang and why is it important and does it influence the choice here?
Response 14: The research of Wang Chunchao et al. (2018) showed that cognitive ability has a significant positive impact on the wage income of workers. And further mechanism analysis shows that non-cognitive ability promotes the increase of workers' wage income through social capital effect, occupational screening effect and education marginal effect.
This document is used select the mediating variable in this study because it has certain similarities and references in terms of variable measurement methods, research neighborhoods, and influence mechanisms. First of all, this study is also based on the “Big Five“ personality measures of non-cognitive ability. In addition, this study studies the influence of non-cognitive ability on micro-individual behavior and decision-making results like this study, and has a certain similarity in the impact mechanism. Therefore, this study considers this literature as the primary reference when selecting mediating variables.
Round 2
Reviewer 3 Report
The authors have corrected the manuscript according our remarks.
Abstract has now been accomplished.
The authors have completed the definition of NCA and the manuscritp is well improved.
Please consider these elements to correct the manuscripts:
"China has been entering a particular period of green production transformation since 2005." Please can you indicate a source/reference
Mediating Variables and Control Variables
"... this study studies" please modify with this study explores or another terms.
Author Response
Response to Reviewer 3 Comments
Point 1: Comments and Suggestions for Authors:
The authors have corrected the manuscript according our remarks. Abstract has now been accomplished. The authors have completed the definition of NCA and the manuscript is well improved. Please consider these elements to correct the manuscripts:
Response 1: The author is very grateful to the reviewer 3 for your valuable comments on the abstract, introduction, the definition of NCA, etc of this article in the first round of revisions. Under the guidance of the expert, the quality of this article has also been improved.
Point 2: "China has been entering a particular period of green production transformation since 2005." Please can you indicate a source/reference
Response 2: The author supplements the above statement in the literature. For details, see reference 1.
Point 3: Mediating Variables and Control Variables."... this study studies" please modify with this study explores or another terms.
Response 3: Approved the opinions of the reviewer, which have been revised to “ this study explores the influence of non-cognitive ability on micro-individual behavior and decision-making results like this study…”.
Finally, thanks again to the reviewer for your very valuable revisions to this article.